# Magnetic/Polyetherimide-Acrylonitrile Composite Nanofibers for Nickel Ion Removal from Aqueous Solution

**DOI:** 10.3390/membranes11010050

**Published:** 2021-01-12

**Authors:** Muhammad Omer Aijaz, Mohammad Rezaul Karim, Hamad F. Alharbi, Nabeel H. Alharthi, Fahad S. Al-Mubaddel, Hany S. Abdo

**Affiliations:** 1Center of Excellence for Research in Engineering Materials (CEREM), King Saud University, Riyadh 11421, Saudi Arabia; 2Advanced Membrane Technology Research Centre (AMTEC), School of Chemical and Energy Engineering (SCEE), Faculty of Engineering, Universiti Teknologi Malaysia (UTM), Skudai, Johor 81310, Malaysia; 3K.A. CARE Energy Research and Innovation Center, Riyadh 11451, Saudi Arabia; 4Mechanical Engineering Department, King Saud University, Riyadh 11421, Saudi Arabia; harbihf@ksu.edu.sa (H.F.A.); alharthy@ksu.edu.sa (N.H.A.); 5Chemical Engineering Department, King Saud University, Riyadh 11421, Saudi Arabia; falmubaddel@ksu.edu.sa; 6Mechanical Design and Materials Department, Faculty of Energy Engineering, Aswan University, Aswan 81521, Egypt; habdo@ksu.edu.sa

**Keywords:** polyetherimide, polyacrylonitrile, nanofiber membranes, magnetic nanoparticles, nickel ions

## Abstract

In this study, a magnetic/polyetherimide-acrylonitrile composite nanofiber membrane with effective adsorption of nickel ions in an aqueous solution was created using a simple electrospinning method. Iron oxide nanoparticles (NPs) were stirred and ultrasonically dispersed into a polyetherimide-acrylonitrile solution to create a homogenous NPs suspension, which was placed in an electrospinning machine to produce a uniform and smooth nanofiber composite membrane. Nanoparticle incorporation into this membrane was confirmed using scanning electron microscope, energy dispersive X-ray spectroscopy (EDX), Fourier-transform infrared spectroscopy (FTIR), X-ray diffraction (XRD), and NPs aqueous stability from a leaching test. The high adsorption capability of the membrane on nickel ions was attributed to the combination of magnetic NPs, polyetherimide-acrylonitrile matrix, and the nanostructure of the membrane. A membrane containing magnetic NPs demonstrated the maximum adsorption capabilities (102 mg/g) of nickel ions in an aqueous solution. Various kinetic and isotherm models were applied to understand the adsorption behavior, such as pseudo-second-order kinetic and Langmuir isotherm models. A polyetherimide-acrylonitrile composite nanofiber membrane containing magnetic NPs could be used as an environmentally friendly and nontoxic adsorbent for the removal of nickel ions in an aqueous medium due to its ease of preparation and use and stability in aqueous mediums.

## 1. Introduction

Metal ions are a major contaminate of water resources and create numerous problems for environmental ecosystems. Nickel ions (${\mathrm{Ni}}^{2+}$) are a common environmental toxicant that are widely produced by various industries and products, such as lead framing, electroplating, tableware, plastics manufacturing, fertilizers, nickel-based batteries, pigments, metal finishing, connecters, mining, and metallurgical operations. Nickel is a carcinogen because its ingestion in high concentrations can create numerous health problems, such as kidney damage and lung and stomach problems (e.g., vomiting, diarrhea, renal edema, skin dermatitis, nausea, and pulmonary fibrosis) [1,2,3].

To address this issue, much effort has been devoted to developing effective and inexpensive methods for removing ${\mathrm{Ni}}^{2+}$, as well as to the treatment of metal ions in water resources, such as chemical precipitation, ion-exchange, solvent extraction, reverse osmosis, filtration, evaporation, and adsorption [4,5,6]. The removal of metal ions from water using the adsorption method is popular because of its low cost, ease of operation, high efficiency, and low level of pollution [7,8,9]. Many adsorbents have been successfully used to adsorb pollutants, such as biomaterials, carbon materials, metal oxide nanoparticles (NPs), and nanofibrous membranes. Nanofiber membranes attract the most attention among researchers because of their high specific area and porosities that allow them to capture more metal ions on their free sites [10,11,12,13]. The electrospinning method is able to produce different compositions of electrospun nanofibers with enormous surface areas, high porosity, and gas permeability [14,15].

Polyacrylonitrile (PAN) is a semicrystalline synthetic polymer with a tremendous ability to form nanofiber sheets with chemical stability [16,17]. PAN has the capability to produce numerous products, such as ultrafiltration mats, hollow fibers for reverse osmosis, and fibers for textiles. PAN tends to swell and plasticize in aqueous mediums due to good hydrophilicity [18]. To overcome this problem, an amorphous material, polyetherimide (PEI), which has an exceptional film-forming ability, in addition to a superior thermal, mechanical, and chemical stability, was introduced. The successful blending of PEI and PAN improved not only the smoothness of the nanofiber membrane but also improved the thermal properties of the PAN [19].

Inorganic NP adsorbents, such as metal oxides, carbon nanotubes, and polymeric nanoadsorbents, are suitable nanomaterials for removing heavy metals from an aqueous system because of their unique chemical and physical properties [20,21,22,23,24,25]. These NPs agglomerate easily due to their low solubility in aqueous mediums and are difficult to recycle. Iron oxide NPs are more attractive for use as an adsorbent due to their unique features, such as small size, high surface to volume ratio, and magnetic properties [26,27,28,29,30,31]. Therefore, mixing iron oxide NPs with electrospun polymeric nanofibers can overcome the agglomeration and recycling problem, and the mixture can then be used as an efficient adsorbent for the removal of heavy metal ions [32,33,34].

PEI-based nanofiber membranes functionalized with iron oxide have not yet been studied for their adsorption of heavy metal ions from an aqueous system, which the primary novelty of this study. The primary objectives were to modify and characterize the PEI-based composite nanofiber membrane and determine its feasibility as an efficient adsorbent for removal of ${\mathrm{Ni}}^{2+}$ from the aqueous solution. Furthermore, to evaluate the adsorption performance, different adsorption conditions were investigated, such as pH, time, and initial concentration. Isotherms, kinetic models, energy dispersive X-ray spectroscopy (EDX), Fourier-transform infrared spectroscopy (FTIR), X-ray diffraction (XRD), and Brunauer Emmett Teller (BET) theory were investigated.

## 2. Materials and Methods

(a) Adsorbent core materials: PAN (MW = 150,000), PEI (ULTEM™ Resin 1000, SABIC, Riyadh, Saudi Arabia), and magnetic nanoparticles (Iron(III) oxide (Fe_2_O_3_) (Sigma Aldrich)); (b) solvent for electrospinning solution N, N-Dimethylformamide (DMF) (Sigma Aldrich); (c) for adjusting pH: hydrochloric acid (HCl) ) (Sigma Aldrich) and sodium hydroxide (NaOH) (Sigma Aldrich); (d) metal ions for adsorption study: nickel(II) acetate tetrahydrate ($C_{4}H_{14}{\mathrm{NiO}}_{8}$) (Sigma Aldrich). Chemicals were analytical grade, and no modification was performed prior to use.

### 2.1. Solutions for Electrospinning Machine

To prepare the homogenous PEI and PAN solutions, 2 and 10% concentrated PEI and PAN were added separately to DMF solvent and stirred for 60 min at 80 °C and room temperature, respectively. The PEI and PAN solutions were then mixed homogeneously together at a 3:1 ratio by using a magnetic stirrer at 80 °C for 60 min and labeled as the PEI-acrylonitrile nanofiber membrane (PEI-AN) solution. Next, 2% iron oxide was introduced to the solution and mixed for 30 min at a temperature of 80 °C. To improve magnetic nanoparticle dispersion and remove bubbles from the solution, sonication was performed for 60 min and the solution was then labeled as PEI-acrylonitrile with magnetic particle (Fe_2_O_3_) nanofiber membrane (Fe/PEI-AN).

### 2.2. Production of Electrospun Nanofibers

The PEI-AN and Fe/PEI-AN solutions were used to produce nanofibers in an electrospinning machine with a 30 kV maximum capacity (MECC, Fukuoka, Japan; model NF-500). A 10 mL syringe with a needle diameter of 0.6 mm was filled with the PEI-AN and Fe/PEI-AN solutions for electrospinning separately, then the syringe was attached to a pump assembly and the needle was connected to a high voltage supply (Scheme 1). The optimized electrospinning parameters were reported in a previous study [19]. The applied voltage was 19 kV, flowrate was 0.6 mL/h, needle to collector distance was 15 cm, and humidity was 20%. After collecting the nanofiber membranes from the drum collector, the membranes were placed in an oven to dry at 60 °C for 24 h and the composite adsorbents were labeled as PEI-AN and Fe/PEI-AN.

### 2.3. Characterizations

#### 2.3.1. Morphological Study

The morphologies of PEI-AN and Fe/PEI-AN composite nanofibers were performed using a field emission scanning electron microscope (FE-SEM (JSM-7600, JEOL, Japan). In the FE-SEM chamber, a small piece of composite nanofibers sputtered with a layer of platinum for 60 s in a vacuum was used. The average diameters of 50 individual fibers were calculated using the JSM-7600 software.

#### 2.3.2. Infrared Spectra Analyses

FTIR (VERTEX-70, Bruker) was used in the range of 600–4000 cm^−1^ to identify the functional groups of PEI-AN and Fe/PEI-AN composite nanofibers.

#### 2.3.3. X-ray Diffraction

XRD (D8 Discover, Bruker) measurements were performed on PEI-AN and Fe/PEI-AN composite nanofibers to confirm the compositions and metal ion attachment on the composite adsorbent before and after adsorption. An XRD test was conducted using a Cu Kα pulse lamp operated at 40 kV/40 mA at an angle range of 5°–100° with a scanning speed of 2°/min.

#### 2.3.4. Surface Area Measurements (BET)

The specific surface area, pore diameter, and pore volume of the composite nanofiber adsorbents were measured using surface area and pore volume analysis (FlowPrep 060, Micromeritics Instrument Corp., Norcross, GA, USA).

#### 2.3.5. Adsorption Study

The stock solution of ${\mathrm{Ni}}^{2+}$ for adsorption was prepared by adding 0.42 g of $C_{4}H_{14}{\mathrm{NiO}}_{8}$ into 1000 mL of deionized water. A certain amount of composite nanofibers were immersed in the ${\mathrm{Ni}}^{2+}$ solution and shaken at 25 °C. The resulting concentration of the ${\mathrm{Ni}}^{2+}$ solution after adsorption was measured using Atomic Absorption Spectroscopy (AAS). The adsorption capacity and removal percentage of metal ions were calculated using the below equations.
$$Adsorption\;Capacity\;\left(Q\right)=\;\frac{C_{0}-C_{e}}{M}*V$$
$$Removal\;\left(\%\right)=\;\frac{C_{0}-C_{e}}{C_{0}}*100$$
where *Q* is the amount of metal ions adsorbed in milligrams (mg/g); $C_{0}$ and $C_{e}$ are the initial and final concentrations, respectively, in parts per million (ppm), *V* is the volume of the metal ion solution in liters (L), and *M* is the mass of the adsorbent in grams (g). Reported adsorption data were the average of triplet values calculated with the same parameters.

To study the influence of pH (2–8), time (1–180 min), and concentration (50–450 ppm) values on the adsorption of ${\mathrm{Ni}}^{2+}$, approximately 12 mg of composite nanofibers were added to 10 mL of ${\mathrm{Ni}}^{2+}$ solution while shaking at 400 rpm. The pH values were adjusted using 0.5 HCl and 0.5 NaOH, and the effects of pH at higher values (pH ≥ 8) were not recorded due to the precipitation produced by the metal hydroxide (Ni precipitation process occurs in the range of 8.5 to 10.5 pH value) [35,36,37,38].

## 3. Results

### 3.1. Morphological Study

Figure 1 shows the morphological and digital evidence of the prepared composite nanofibers with and without magnetic particles. Figure 1a shows the smooth, bead-free, and drop-less fine fibers of PEI-AN nanofibers with an average diameter of 230 nm; the zoomed-in image shows stable fibers with a wrinkled and creased surface that could assist in seizing metal ions. Figure 1b shows the morphological SEM images of the Fe/PEI-AN membrane, where iron particles fused with fiber can easily be seen. EDX results further confirm the presence of iron (1.7%) in the nanofibers shown in Figure 1c. Digital images of the prepared membranes are shown in Figure 1d; the dark color confirms the addition of Fe_2_O_3_ in the Fe/PEI-AN membrane. Further presence of iron embedded in the membranes is confirmed by the magnetic attraction in Figure 1e.

### 3.2. Leaching Test of Magnetic Particles

To investigate the stability of Fe_2_O_3_ NP in Fe/PEI-AN composite nanofiber, 30 mg of Fe/PEI-AN was added to 25 mL of distilled water for 60 h at 400 rpm. The amount of NP that leached out into water was measured using an inductivity coupled plasma optical emission spectrophotometer (PerkinElmer Optima 4300 DV, Wellesley, MA, USA). No traces were found in the water that confirmed sufficient incorporation of Fe_2_O_3_ NP in the Fe/PEI-AN membrane. Therefore, a composite nanofiber containing magnetic particles can be used as an efficient material for the adsorption of metal ions with no further contamination.

### 3.3. Infrared Spectra Analyses

Figure 2 shows the ATR-FTIR spectral curves to investigate the chemical structure of PEI-AN and Fe/PEI-AN composite nanofibers. The presence of PEI and PAN in both the curves was confirmed by the characteristic nitrile group (C≡N) of PAN at 2245 cm^−1^ and imide group (R-CO-N-CO-R’) of PEI at 1774 and 1721 cm^−1^ [19]. The presence of iron oxide in the Fe/PEI-AN nanofibers was further confirmed by the characteristic peak at 575 cm^−1^, attributed to the stretching vibrations (Fe–O) of iron oxide NP [39]. The identical peaks confirmed the presence of PEI, PAN, and magnetic NP in the composite nanofibers.

### 3.4. X-ray Diffraction

X-ray diffractograms of PEI-AN and Fe/PEI-AN are shown in Figure 3. In Figure 3a, two typical crystalline peaks of PAN appear at 16.6° and 29.2°, which reflects the linear structure of PAN, matching with crystal plane at 100 (16.6°) and 110 (29.2°) [40]. No peaks were recorded for PEI because of its amorphous nature. Figure 3b presents the diffraction peaks related to iron oxide at 35.7°, 43.34°, 53.8°, 57.1°, and 62.8°. Wherein peaks at 35.7° (110) and 53.8° (116) are attributed to a hematite phase (α-Fe_2_O_3_) while peaks at 43.34° (400), 57.1° (511), and 62.8° (440) present the maghemite phase of iron oxide (ɣ-Fe_2_O_3_) [41] powder diffraction file ID (PDF: 01-078-6916). These attributed peaks further confirmed the presence of Fe_2_O_3_ NPs in the composite nanofibers.

### 3.5. Metal Ion Adsorption Study

To determine the maximum adsorption capacity, the stepwise effects of different conditions, such as pH values, time intervals, and initial concentrations, were analyzed. Adsorption of prepared adsorbents for ${\mathrm{Ni}}^{2+}$ were first measured on the basis of pH values ranging from 2 to 8. Experiments were performed using 10 mL of initial ${\mathrm{Ni}}^{2+}$ at a concentration of 25 mg/L, as well as 12 mg of PEI-AN and Fe/PEI-AN nanofibers at room temperature. Figure 4a shows the removal percentage of ${\mathrm{Ni}}^{2+}$ in the pH range of 2–8. The adsorption percentage increased with increasing pH and reached its highest values at approximately pH ≤ 8. Lower adsorption percentages were observed at lower pH values due to the formation of H^+^ that covered the metal ions [42,43,44]. The PEI-AN and Fe/PEI-AN showed maximum adsorption efficiencies of ${\mathrm{Ni}}^{2+}$ at a pH ≤ 8 that were 83 and 91%, respectively.

Figure 4b shows the effect of adsorption time on the adsorption capacity of PEI-AN and Fe/PEI-AN composite nanofibers. The time study was performed using 10 mL of initial ${\mathrm{Ni}}^{2+}$ at a concentration of 25 mg/L, as well as 12 mg of PEI-AN and Fe/PEI-AN at a pH ≤ 8 for 1–180 min. The adsorption capacity sharply increased as time increased to 30 min. After 30 min, the adsorption growth rate gradually decreased and reached an adsorption equilibrium at 60 min. A sharply increased adsorption rate (until 30 min) might have been due to the availability of free sites, more surface area, and porosities of nanofiber adsorbents. After 30 min, the growth rate of the adsorption capacity decreased with increasing contact time until an adsorption equilibrium was reached. This phenomenon was due to the limited availability of adsorptive sites and metal ion concentration. A similar effect of time on adsorption was seen for both adsorbents, but the iron-containing adsorbent showed a higher adsorption capacity. This might have been due to the presence of iron particles that enhanced the specific surface area and adsorption free active sites.

Figure 4c shows the effect of the initial concentration of the adsorption of ${\mathrm{Ni}}^{2+}$ onto PEI-AN and Fe/PEI-AN composite nanofibers. For the concentration experiments, 10 mL of initial ${\mathrm{Ni}}^{2+}$ concentration (50–450 mg/L) and 12 mg of nanofiber adsorbent were used to determine the maximum adsorption of metal ions at pH ≤ 8 and 60 min. As the initial concentration of ${\mathrm{Ni}}^{2+}$ increased, the adsorption capacity improved gradually, and Fe/PEI-AN particularly exhibited a higher ${\mathrm{Ni}}^{2+}$ adsorption capacity than PEI-AN. This increased adsorption of Fe/PEI-AN was due to the availability of more adsorption sites for ${\mathrm{Ni}}^{2+}$ provided by NPs inside the adsorbent [45]. The amount of adsorption capacity was increased sharply as the initial concentration changed from 50 to 250 mg/L. After 250 mg/L of initial concentration, the adsorption tendency became slow and reached an equilibrium value of approximately 84 mg/g (PEI-AN) and approximately 102 mg/g (Fe/PEI-AN) when the initial concentration was 350 mg/L. This concentration study also demonstrated the dominant effect and better efficiency of the absorption performance of Fe/PEI-AN for ${\mathrm{Ni}}^{2+}$ compared with the PEI-AN adsorbent. The Fe-O functional groups of iron oxides could be responsible for the increased adsorption rate of ${\mathrm{Ni}}^{2+}$ on the Fe/PEI-AN adsorbent.

### 3.6. Kinetics and Isotherm Models for Adsorption Study

To better explore the adsorption mechanism of composite nanofibers during the process of adsorption, various kinetic models were applied, such as pseudo-first-order (PFO), pseudo-second-order (PSO), Elovich, power function, and intraparticle diffusion. To determine the closeness between the experimental and model-predicted adsorption data, the standard error of estimate (SEE) and the coefficient of determination (R^2^) were calculated. Linear expressions of the above-mentioned kinetics and estimations are expressed in Table 1 and Table 2, respectively.

Figure 5 shows the plotted kinetic models and Table 3 indicates the calculated parameters along with the R^2^ and SEE values of PFO, PSO, Elovich, power function, and intraparticle diffusion. Both composite nanofibers showed an R^2^ value of PSO closer to 1, revealing that PSO was more suitable than PFO for both adsorbents; PSO indicated that the adsorption process was chemisorption and exchange or share of electrons between adsorbent and adsorbate [52]. The high value of the Elovich *α* parameter further confirmed the chemisorption nature of the adsorbents [53,54]. The high value of the *c* parameter of intraparticle diffusion indicated the adsorption process also had a boundary layer effect. The rate coefficient (*k_f_*) value as estimated from the power function model was higher for Fe/PEI-AN, suggesting an increased adsorption of ${\mathrm{Ni}}^{2+}$ with time.

The ${\mathrm{Ni}}^{2+}$ adsorption equilibrium data of PEI-AN and Fe/PEI-AN composite nanofibers retrieved from the concentration study were analyzed by Langmuir, Freundlich, and Temkin isotherm models. The equations of these models are listed in Table 1. The fitted lines of the applied models and a summary of calculated parameters along with error functions are shown in Figure 6 and Table 4.

According to the calculated parameters, R^2^ values of both adsorbents showed the best fit with the Langmuir model (R^2^ > 0.96), and Fe/PEI-AN particularly showed a high value of R^2^ (approximately 0.98) with close SEE and ∆*q* values. Compared with the Langmuir model, the Temkin model values suggested that the amount of adsorption had a minor impact on heat. High values of R^2^ and close agreement of the theoretical adsorption capacity with the experimental values in the Langmuir model suggested a monolayer adsorption of ${\mathrm{Ni}}^{2+}$ onto the surface of composite nanofibers.

### 3.7. Adsorption Mechanism Characterizations

After the adsorption study, characterizations, such as EDX with mapping, FTIR, and XRD, were performed on PEI-AN and Fe/PEI-AN adsorbent to confirm the bonding of ${\mathrm{Ni}}^{2+}$ ions onto composite nanofibers. EDX mapping and spectra confirm the presence of ${\mathrm{Ni}}^{2+}$ onto the PEI-AN and Fe/PEI-AN adsorbents, as shown in Figure 7 and Figure 8, respectively. Figure 9 shows the FTIR spectral curves for adsorbents after adsorption studies that provide further evidence for the successful attachment of ${\mathrm{Ni}}^{2+}$ onto the composite nanofibers. Figure 9a,b shows the FTIR curves of adsorbents after adsorption, in which new peaks related to metal oxide vibration are found at 580 and 670 cm^−1^ (curve a) and 629 and 590 cm^−1^ (curve b); these peaks could be assigned to a Ni-OH stretching bond [55,56,57]. After adsorption (curve b) the characteristic peak at 575 cm^−1^ that caused the stretching vibrations (Fe–O) produced a new peak at 590 cm^−1^ due to the intrinsic Fe-O interaction with ${\mathrm{Ni}}^{2+}$ [58,59]. These results confirmed that the ${\mathrm{Ni}}^{2+}$ was adsorbed into the adsorbents and interacted with magnetic particles (Fe_2_O_3_). In addition, Figure 10 shows the XRD results that suggest ${\mathrm{Ni}}^{2+}$ bonding with composite nanofibers by presenting two new diffraction peaks at 8.9° and 9.4°. These peaks were likely due to the reflection of adsorption layered on adsorbents after the adsorption of ${\mathrm{Ni}}^{2+}$ [60].

The porous structure and surface area of the composite nanofibers were also analyzed using the BET method to show the effect of porosity and surface area on adsorption capacity. The surface area, pore volume, and average pore diameter recorded in Table 5 show higher values of Fe/PEI-AN compared with PEI-AN composite nanofibers. The higher porosity and surface area confirmed that Fe/PEI-AN had more adsorption sites for ${\mathrm{Ni}}^{2+}$. In addition to higher porosity and surface area, the other ${\mathrm{Ni}}^{2+}$ adsorption mechanism of Fe/PEI-AN composite nanofibers is shown in Scheme 1. Fe_2_O_3_ was involved in the adsorption mechanism from an ion-exchange reaction between the proton and ${\mathrm{Ni}}^{2+}$.

Before concluding, it is important to show how a recently developed PEI-AN composites nanofiber was used with magnetic NP that did not leach into the aqueous medium. This composite nanofiber adsorbent was successfully applied for the adsorption of metal ions, resulting in a higher adsorption capacity compared with the various adsorbents listed in Table 6. Additionally, to evaluate the practical use of developed Fe/PEI-AN composite adsorbent, Figure 11 shows the preliminary adsorption study results of different metal ions (i.e., Cd, Cu and Ni ions). We hope that these results will assist the research community in future work.

This study successfully used electrospinning to produced novel nanocomposite nanofiber adsorbents for ${\mathrm{Ni}}^{2+}$ removal. The adsorption process demonstrated that a Fe/PEI-AN composite nanofiber membrane showed maximum adsorption capacity (102 mg/g) at an optimized pH value of ≤8, a time interval of 60 min, and an initial concentration of 350 mg/L. For a possible explanation of the adsorption mechanism, the collected adsorption data were correlated with conventional adsorption kinetics and isotherm models. Among the conventional kinetic models, PSO kinetic models fitted well compared with other models, suggesting that a vital chemical reaction was involved during the ${\mathrm{Ni}}^{2+}$ adsorption process. Moreover, the Langmuir isotherm model defined the representation of the adsorption data well compared with the Freundlich and Temkin models and the proposed monolayer adsorption was dominant during ${\mathrm{Ni}}^{2+}$ adsorption. The applied models and isotherms were further confirmed through the adsorption mechanism tests of EDX, FTIR, XRD, and BET. These investigations proposed that iron oxide NPs in the composite nanofiber membranes enhanced ion exchange and improved adsorption sites during adsorption of ${\mathrm{Ni}}^{2+}.$ The improved adsorption capacity of Fe/PEI-AN (approximately 21%) was primarily due to the increased porosity, surface area, and availability of Fe-O functional groups of iron oxide. Developed novel composite membranes may have numerous applications where environmentally friendly and nontoxic adsorbents for the removal of ${\mathrm{Ni}}^{2+}$ in an aqueous medium are required.

## Data Availability

The data presented in this study are available on request from the corresponding author.

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
