# Peer review of "Magnetic/Polyetherimide-Acrylonitrile Composite Nanofibers for Nickel Ion Removal from Aqueous Solution"

_membranes, 2021, doi:10.3390/membranes11010050_

Round 1

Reviewer 1 Report

The authors report the preparation and removal properties of magnetic/polyetherimide-acrylonitrile composite nanofibers. This study is interesting and publication is recommended after major revision.

  1. Does this composite only adsorb Ni ions? How about Co, Cd, and Hg ions?
  2. This research reminded me that some crystalline materials (or composites) also displayed interesting adsorption or removal properties for toxic metals, such as reports in Chem. Eur. J. 2018, 24, 9729.; ACS Appl. Nano Mater. 2020, 3, 9724. Those references should be cited.

Author Response

Reviewer_1

Comments and Suggestions for Authors

The authors report the preparation and removal properties of magnetic/polyetherimide-acrylonitrile composite nanofibers. This study is interesting and publication is recommended after major revision.

1. Does this composite only adsorb Ni ions? How about Co, Cd, and Hg ions?

Answer: Fe/Pei-AN is a novel composite that never used for any application. To check the adsorption properties of metal ion first we started from Ni ion. After successfully applied novel composite adsorbent for the removal of Ni Ions. Now we have considered to check the removal of further metal ion (i.e. Co, Cd, and Hg ions) in our upcoming works.

2. This research reminded me that some crystalline materials (or composites) also displayed interesting adsorption or removal properties for toxic metals, such as reports in Chem. Eur. J. 2018, 24, 9729.; ACS Appl. Nano Mater. 2020, 3, 9724. Those references should be cited.

Answer: As per suggestion, recommended article has been cited and highlighted in manuscript as reference no. 24)

Reviewer 2 Report

The manuscript “Magnetic/Polyetherimide-Acrylonitrile Composite Nanofibers for Nickel Ions Removal from Aqueous Solution” describes the experimental results of composite adsorbent preparation and their adsorption properties towards Ni(II) ions.

Main comments:

(i) The obtained nanocomposite has to characterized as membrane (water permeability, porosity etc.). In other case, term “membrane” must be withdraw from text or changed on “adsorbent”.

(ii) What was the reason for choosing the nickel acetate for adsorption experiments? Because, the real wastewaters as role content nitrate, sulfate or chloride salts.

(iii) The pHzpc value of composite should be measured for discussion of the pH influencing (doi:10.1016/j.cis.2012.01.005, http://dx.doi.org/10.1016/j.watres.2017.04.014). Also, the study of adsorption under pH>7 is unreasonable, due to Ni(OH)2 precipitation.

(iv) Fe2O3  has four polymorphs: α-Fe2O3 (hematite), β-Fe2O3, γ-Fe2O3 (maghemite) and ε-Fe2O3. XRD data discussion should be revised for identification of polymorphic modification.

(v) The results of kinetic studies should be compared with others adsorbents (https://doi.org/10.1016/j.chemosphere.2016.12.062), as well as for understanding the advantages of prepared composite the Table 6 should be revised taking into account high performance adsorbents (https://doi.org/10.1016/j.carbpol.2019.01.045).

Minor comments:

(i) The obtained adsorbent has a very low specific surface area and providing of BET analysis data non-informative (DOI 10.1515/pac-2014-1117).

(ii) References related to isotherm and kinetic modeling should be changed of review manuscripts (http://dx.doi.org/10.1016/j.jtice.2017.01.024, doi:10.1016/j.cej.2009.09.013).

The manuscript needs major revisions.

Author Response

Reviewer_2

Comments and Suggestions for Authors

The manuscript “Magnetic/Polyetherimide-Acrylonitrile Composite Nanofibers for Nickel Ions Removal from Aqueous Solution” describes the experimental results of composite adsorbent preparation and their adsorption properties towards Ni(II) ions.

 Main comments:

  • The obtained nanocomposite has to characterized as membrane (water permeability, porosity etc.). In other case, term “membrane” must be withdraw from text or changed on “adsorbent”.

Answer: Thanks for suggestion, we have added the words “adsorbent” and “composites nanofiber” in the manuscript as required and highlighted in manuscript.

  • What was the reason for choosing the nickel acetate for adsorption experiments? Because, the real wastewaters as role content nitrate, sulfate or chloride salts.

Answer: Fe/Pei-AN is a novel composite that never used for any application. To check the adsorption properties of metal ion first we started from Ni ion adsorption study. After successfully applied novel composite adsorbent for the removal of Ni Ions. Now we have planned to check the removal of further content from wastewater in our upcoming works.

  • The pHzpc value of composite should be measured for discussion of the pH influencing (doi:10.1016/j.cis.2012.01.005, http://dx.doi.org/10.1016/j.watres.2017.04.014). Also, the study of adsorption under pH>7 is unreasonable, due to Ni(OH)2 precipitation.

Answer: As per literature review, different metal hydroxide is started precipitation at a certain pH range. According to the different studies pH for hydroxide precipitation process is pH 10-10.5 for nickel, pH 9.0-9.5 for zinc and pH 8.5-9.5 for copper [37]. Therefore, we selected pH≤8 to avoid precipitation and more refences added in the manuscript accordingly, also symbol “≤" replaced where "pH of 8" into pH ≤ 8 to avoid misunderstanding for readers.

To fulfill pHzpc requirements, further experiments are needed but due to pandemic situation further experiments are not possible as the lab and other facilities are closed. So, more citations added to support the pH influence data [36-37].

  • Fe2Ohas four polymorphs: α-Fe2O3 (hematite), β-Fe2O3, γ-Fe2O3 (maghemite) and ε-Fe2O3. XRD data discussion should be revised for identification of polymorphic modification.

Answer: As per suggestion, revised description has been added in lines between 188-196 and highlighted in the manuscript.

  • The results of kinetic studies should be compared with others adsorbents (https://doi.org/10.1016/j.chemosphere.2016.12.062), as well as for understanding the advantages of prepared composite the Table 6 should be revised taking into account high performance adsorbents (https://doi.org/10.1016/j.carbpol.2019.01.045).

Answer: As per suggestion, more research studies have been added in table 6 and highlighted in the manuscript

  • The obtained adsorbent has a very low specific surface area and providing of BET analysis data non-informative (DOI 10.1515/pac-2014-1117).

        Answer: As per suggestion, it has been corrected.

  • References related to isotherm and kinetic modeling should be changed of review manuscripts (http://dx.doi.org/10.1016/j.jtice.2017.01.024, doi:10.1016/j.cej.2009.09.013.

Answer: As per suggestion, recommended articles have been cited in the        manuscript and highlighted in the manuscript as reference no. 46-47

Round 2

Reviewer 1 Report

For publication in a high-quality journal such as Membranes, experimental data have to be displayed completely as possible. Therefore, preliminary tests of the title composite for common toxic metals are suggested. The authors cited the wrong papers (see ref. 24). For wide readers, journals reporting other materials or composites with similar properties to that in this manuscript should be also cited.

Author Response

Reviewer_1

Comments and Suggestions for Authors

For publication in a high-quality journal such as Membranes, experimental data have to be displayed completely as possible. Therefore, preliminary tests of the title composite for common toxic metals are suggested. The authors cited the wrong papers (see ref. 24). For wide readers, journals reporting other materials or composites with similar properties to that in this manuscript should be also cited.

Answer: As per suggestion, recommended articles have been cited and corrected as well as highlighted in the manuscript (Ref. from 20-25).

Reviewer 2 Report

The revised manuscript could be reccomended for the publicaion.

Author Response

Reviewer_2

Comments and Suggestions for Authors

The revised manuscript could be reccomended for the publication

Answer: We sincerely appreciate reviewer efforts and recommendation to publish our manuscript in this journal.

Round 3

Reviewer 1 Report

This version might be okay for publication in Membranes. However, I strongly suggest the authors include preliminary tests of the title composite for common toxic metals (such as Co, Cd, and Hg cations). It helps to evaluate the practical uses.

Author Response

Reviewer_1

Comments and Suggestions for Authors

This version might be okay for publication in Membranes. However, I strongly suggest the authors include preliminary tests of the title composite for common toxic metals (such as Co, Cd, and Hg cations). It helps to evaluate the practical uses.

Answer: first, we sincerely appreciate reviewer efforts and recommended to publish our work in present form in Membranes. However, to evaluate the practical use of our developed composite adsorbent, we have added preliminary tests results (for Cd, Cu and Ni ions) of the title composite as strongly suggestion by reviewer. The additions are highlighted in the manuscript in lines no. 309 to 312 and additional figure added and highlighted at line no. 318.

This manuscript is a resubmission of an earlier submission. The following is a list of the peer review reports and author responses from that submission.

Round 1

Reviewer 1 Report

The presented work describes a method for the adsorption of nickel ions by magnetic/polyetherimide-acrylonitrile composite nanofibers membrane. The adsorption capacity of the membrane was greatly improved. Different kinetics and isotherm models were applied to understand the adsorption behavior during the adsorption process. However, the novelty is not enough to publish this work in Membranes. The nanoparticles of Fe2O3 is a well-known material. How about the performance when using different nanoparticles such as FeO. Figure 10 indicated new peaks at 8.9 and 9.4 degrees, implying the generation of new phases or instability for Fe2O3 particles after metal removal experiments. More studies such MOF, phosphate, and phosphite for toxic metal removal researches have been cited and discussed in this draft. Overall, the novelty of this work is not enough, and publication is not recommended.

Reviewer 2 Report

The manuscript “Magnetic/Polyetherimide-Acrylonitrile Composite Nanofibers for Nickel Ions Removal from Aqueous Solution” describes the experimental results of composite adsorbent preparation and their adsorption properties towards Ni(II) ions.

Main comments:

(i) the manuscript does not contain novel results

(ii) the results were discussed at a low level and did not meet the requirements of the Membrane

(iii) the advantage of composite materials containing iron oxide is unclear

(iv) It is unclear what role magnetic particles play when using for adsorbent application

(v) the conditions for carrying out the kinetic experiment were chosen incorrectly , since the low initial concentration of nickel ions does not allow achieving equilibrium adsorption

(vi) the obtained adsorbent has a low specific surface area, in addition, the error of the BET method is more than 20% and the data are given incorrectly.